# R-CNN-Based Ship Detection from High Resolution Remote Sensing Imagery

**Shaoming Zhang [1], Ruize Wu [1], Kunyuan Xu [1], Jianmei Wang [1,*] and Weiwei Sun [2]**

[1]   College of Surveying, Mapping and Geo-Informatics, Tongji University, Shanghai 200092, China;
    08053@tongji.edu.cn (S.Z.); 1731992@tongji.edu.cn (R.W.); 1633308@tongji.edu.cn (K.X.)

[2]   Department of Geography and Spatial Information Techniques, Ningbo University, Ningbo 315211,
    Zhejiang, China; sunweiwei@nbu.edu.cn

[*]   Correspondence: jianmeiw@tongji.edu.cn; Tel.: +86-136-4169-2941

**Abstract:** Offshore and inland river ship detection has been studied on both synthetic aperture radar (SAR) and optical remote sensing imagery. However, the classic ship detection methods based on SAR images can cause a high false alarm ratio and be influenced by the sea surface model, especially on inland rivers and in offshore areas. The classic detection methods based on optical images do not perform well on small and gathering ships. This paper adopts the idea of deep networks and presents a fast regional-based convolutional neural network (R-CNN) method to detect ships from high-resolution remote sensing imagery. First, we choose GaoFen-2 optical remote sensing images with a resolution of 1 m and preprocess the images with a support vector machine (SVM) to divide the large detection area into small regions of interest (ROI) that may contain ships. Then, we apply ship detection algorithms based on a region-based convolutional neural network (R-CNN) on ROI images. To improve the detection result of small and gathering ships, we adopt an effective target detection framework, Faster-RCNN, and improve the structure of its original convolutional neural network (CNN), VGG16, by using multiresolution convolutional features and performing ROI pooling on a larger feature map in a region proposal network (RPN). Finally, we compare the most effective classic ship detection method, the deformable part model (DPM), another two widely used target detection frameworks, the single shot multibox detector (SSD) and YOLOv2, the original VGG16-based Faster-RCNN, and our improved Faster-RCNN. Experimental results show that our improved Faster-RCNN method achieves a higher recall and accuracy for small ships and gathering ships. Therefore, it provides a very effective method for offshore and inland river ship detection based on high-resolution remote sensing imagery.

**Keywords:** ship detection; regional convolutional neural network; GaoFen-2 remote sensing image; small ship; gathering ship

---

## 1. Introduction

Ship detection on remote sensing images has a wide range of applications in civil areas and defense security. Ship detection with satellite imagery can provide real-time location information for navigation management control and maritime search and rescue, which guarantees the effectiveness and safety of work at sea and on inland rivers, such as ocean transportation supply. It also contributes to the supervision and construction of important coastal zones and harbors, which promotes the protection of the ecology and sea health, offshore areas, and inland rivers.

Currently, most of the related research works are based on synthetic aperture radar (SAR) images [1–11]. Nevertheless, it is difficult to build and solve a proper statistical model for a complex sea area that is characterized by an uneven surface, inshore surface, inland river surface,

etc. Therefore, these methods can cause false alarms because of obstructions, such as small islands, garbage, and floating objects [12,13].

In view of the above problems, another approach is to use a target detection algorithm based on optical remote sensing images. During the past decades, optical remote sensing images have provided an abundance of shape, outline color, and texture information, and ship detection using 2D object detection algorithms in remote sensing imagery has been extensively studied [14–16]. The classic methods of ship detection are based on threshold segmentation [17], which requires a favorable condition of the sea surface; however, its detection results are not sufficiently satisfactory. Then, many groups of researchers began to use classifiers such as support vector machine (SVM), AdaBoost, decision trees, etc. [18,19], which are based on hand-engineered features such as the local binary pattern (LBP), histogram of oriented gradient (HOG) [20], Gabor and so on [21,22]. In addition, a method based on the mixture of DPMs [23] is able to detect ships close to each other. However, these classic methods are limited by manually designed image features and templates [24–26] and encounter bottlenecks when ships vary in size and position. Recently, object detection algorithms based on machine learning, especially deep learning, have been used in both SAR and optical remote sensing [27–35]. A deep convolutional neural network (DCNN) can extract semantic level image features that are robust to image noise and morphological changes and relative positions of targets [36–39]. The DCNN-based methods make it possible to detect ships with a variety of different sizes, shapes, and colors and achieve a better result than traditional target detection methods. However, most of the studies combine a CNN with SAR images that have no colorful features since it used in optical remote sensing images. Moreover, it is still a challenge to detect small ships and ships that are densely close to each other.

To address the above problem, especially for small ships and ship clusters detection, in this paper, we propose an improved R-CNN method for ship detection on optical remote sensing images. With the continuous progress of remote sensing techniques, a large number of remote sensing imagery with high spatial resolution that is greater than 1 m have become available. Examples of such imagery include Quick Bird, GeoEye, WorldView, GaoFen-2 (GF2), GaoFen-4, and so on. These submeter resolution remote sensing images make it practical to detect small ships and distinguish ships from clusters. In this paper, we focus on GF2 optical images, which are obtained from China's first multispectral remote sensing satellite and have 1-m resolution [40]. The high resolution of GF2 images has plentiful color and texture information and details, which are essential to detect small ships and distinguish ships within clusters. Theoretically, images with higher resolution, such as Quick Bird and GeoEye, can lead to a better detection result. However, it is more economical to use GF2 in our research.

For the main processes, due to the very large area covered by a single remote sensing image, we preprocessed the image with SVM, which classifies the water and nonwater area first. Then, we extract ROIs from nonwater areas using HOG features and morphology analysis. It approximately generates many ROI images that are small images that may contain ships. The most important process is ship detection on these ROI images with the R-CNN-based algorithm, which accurately detects the location of ships. For the detection algorithm, one of the most popular classes of methods is based on the R-CNN, which usually consists of a CNN to extract features, an RPN to locate ships [34], and another class of methods based on SSD [41]. We use a classic R-CNN-based method, Faster-R-CNN [34], and improved the CNN structure and RPN details for small ship and ship cluster detection. Referring to HyperNet [42] and PvaNet [43], we combine features of different resolutions together based on a classic network VGG-16 [44]. Then, we modified the default shape of anchor boxes in the RPN for ship detection. Moreover, with an upsampling operation, we put a larger feature map into the ROI pooling layer after the RPN. Finally, we experimented and compared the performance of the classic DPM method, SSD300, YOLOv2 [45], Faster-R-CNN with ZF-net, the original VGG-16, and our modified VGG-16.

## 2. Ship Detection Method

### 2.1. Ship Detection Process

We adopted a strategy from coarse and whole to fine and partial, in which we roughly look for ROIs that may contain ships first, and then detect every single ship accurately on candidate ROI images. The area covered by each remote sensing image is very large, usually covering a square larger than 20,000 m×20,000 m. Therefore, it is hard to process the whole image at one time. Figure 1 shows the main process, which includes the following steps.

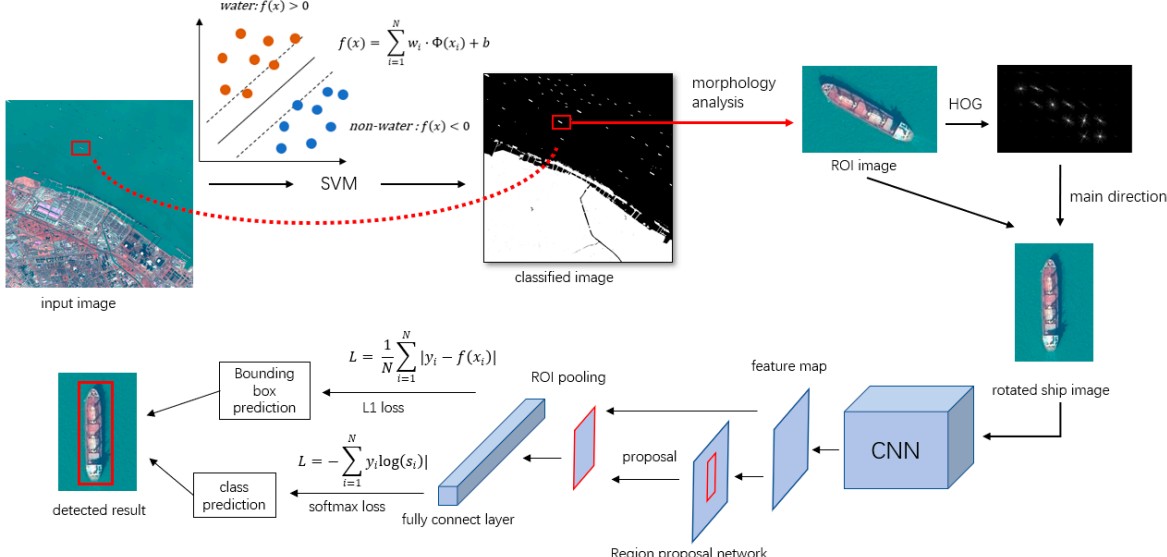

**Figure 1.** Ship detection process.

(1) Given a GF2 image, perform water and nonwater segmentation using SVM. In addition, obtain the ROIs that may contain ships by analyzing the size and aspect ratio of the nonwater area.

(2) Obtain the main direction of ships in the ROI by computing the HOG feature of the ROI image. In addition, rotate the ROI image to the main direction.

(3) Perform the R-CNN- based ship detection method on the ROI image.

### 2.2. Water-Land Segmentation and Ship ROI Extraction

To obtain candidate areas containing ships in the first step, we train a classifier based on SVM to perform water and nonwater segmentation. SVM is known to perform better than other classic algorithms, such as decision trees and Adaboost, and they can achieve good generalization on a small training set; therefore, SVM is suitable for the classification task in the first step.

We choose the radial basis function as the kernel function in SVM and set the penalty parameter to 1.0 and gamma in the kernel function to 0.25. Then, we evenly select small polygons from the water areas (including inland rivers and nearshore water) and nonwater areas (including land and ships). The pixels in these polygons with 4 channel values are labeled as the training set. The polygons are selected on 8 training images, and each image contributes more than 400,000 pixels for each label. Then, we trained SVM to convert the input image to a binary classified image.

Next, the foreground and background were represented by white pixels and black pixels on the binary image. We got many polygons by finding contours of the foreground pixels. As shown in Figure 2b, the area and the minimum bounding rectangles of the polygons were calculated. Then, we filtered out the ROI candidates by logical judgment. Polygons with area between 1000 and 100,000 square meters were retained as gathering ships or huge ships. Polygons with aspect ratio of bounding rectangles between 2.0 and 8.0 and area from 300 to 1000 square meters were retained as

single ships and the other polygons were discarded. The ROI images were cropped from the original remote sensing images with the expanding coordinates of the remained polygons.

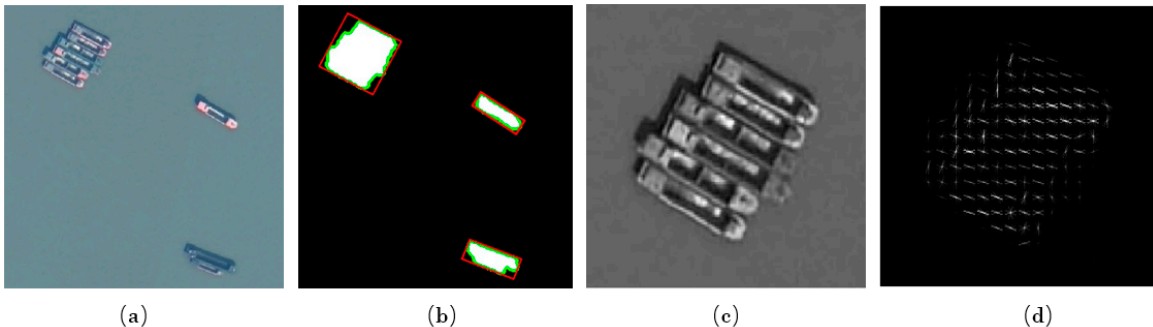

**Figure 2.** (**a**) Part of original image, (**b**) contours result, (**c**,**d**) histogram of oriented gradient (HOG) description.

Considering that only horizontal rectangular boxes are detected in the R-CNN, we preprocess the ship ROIs by rotating them to a main direction, which can be useful while encountering ship clusters. As shown in Figure 2d, the main direction was detected by HOG features with a cell size of 16 × 16 and block size of 1. Sobel was used to calculate the gradient in the x and y directions. We divided 180 degrees to 9 groups, because two directions that were 180 degrees apart were treated as the same. The average gradient direction of the group represented the main direction of the ROI image. The ROI images were rotated to the nearest horizontal/vertical orientation. Finally, the rotated ship ROIs are used as R-CNN input images.

### 2.3. Improve R-CNN and Detecting Ships

On the basis of the rotated ship ROI images mentioned above, the R-CNN method is responsible for accurate ship detection. First, we analyze the structure of the Faster-RCNN model and the function of every part of the framework. Then, we analyze the reason for its drawbacks on small and gathering ships' detection. Finally, we modified the CNN and RPN part of Faster-RCNN to improve the detection result.

### 2.3.1. Improved Faster-RCNN Model

The original Faster-R-CNN framework consists of a deep convolutional neural network (CNN) and a region proposal network (RPN). The CNN extracts the image features and downsamples the input image at the same time. The image's shallow features have a higher resolution and more primary information, while the deeper features have a lower resolution and more abstract information. The RPN uses the output of the last convolutional layer and uses a set of anchor boxes on each pixel to calculate the loss function of classification and regression. The structure of Faster-R-CNN is shown in Figure 3. Regarding the gathering ships and small ships, the target pixels on the deep feature map are very few, which indicates a considerable information loss of small ships. In addition, fewer pixels on the feature map in the RPN can lead to lower location accuracy. In the RPN, the anchor boxes are set to a series of normal shapes. However, in the ship detection task, the ships always have a large aspect ratio. The initial shapes of the anchor boxes in Faster-R-CNN may lead to a slower learning process.

As analyzed above, we aimed to use both shallow and deep convolutional features and set a more similar aspect ratio of anchor boxes in the RPN to improve small and gathering target detection.

The original Faster-R-CNN used VGG-16 as feature extraction CNN. The VGG-16 network has 13 convolutional layers and 5 pooling layers, in which pooling layers are used to downsample the image. Image width and height shrink to half of the input image after every pooling layer, so the output of the last layers in VGG is 1/32 of the input size. The convolutional layers can be divided into 4 groups by 5 pooling layers, and the convolutional layers in each group have the same size (named conv1_1,

conv1_2, conv2_1, etc.). Therefore, a ship with a width less than 32 pixels on the input image can be less than 1 pixel on the last convolutional feature map. This is not conducive to small ship detection.

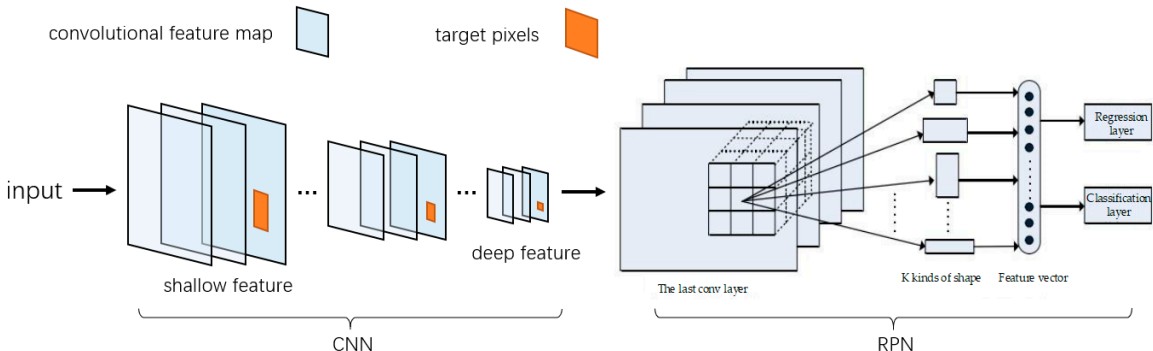

**Figure 3.** Relationship of target, shallow, and deep features in the Faster-R-CNN structure.

We turn to using convolutional features that are both deep enough and larger in size. On the deeper layers, the convolutional features are more abstract and more expressive but lose detailed information. On the lower layers, the features maintain more detailed information such as location and texture information. Therefore, we combine three convolutional feature maps to generate a multiresolution image feature. First, we use the pooling result feature map after the last convolutional layer in the third group, which has the same size of convolutional feature maps in the fourth group. Second, we choose the output of the last convolutional layer in the fourth group, which is 1/16 of the size of the CNN input image. Third, we operate a deconvolutional layer after the deepest convolutional layer so that the output feature map is upsampled to the same size of the maps in the fourth group. Prior operations do not change the channels of maps. Finally, these feature maps of the same size are concatenated by channels, and a convolutional layer with a filter of size 1×1 is used to convert the number of channels to 512. In addition, the multiresolution feature map with a larger size is generated. The new modified VGG structure and the details of the layers are shown in Table 1.

**Table 1.** The architecture of improved VGG.

| Layer | Input | Kernel Size | Stride | Output | Parameters Num | Memory |
|---|---|---|---|---|---|---|
| Conv1_1 | 3@224×224 | 3 × 3 | 1 | 64@224×224 | $64 \times (3 \times 3 \times 3 + 1)$ | $64 \times 224 \times 224$ |
| Conv1_2 | 64@224×224 | 3 × 3 | 1 | 64@224×224 | $64 \times (3 \times 3 \times 64 + 1)$ | $64 \times 224 \times 224$ |
| MaxPool_1 | 64@224×224 | 2 × 2 | 2 | 64@112×112 | 0 | $64 \times 112 \times 112$ |
| Conv2_1 | 64@112×112 | 3 × 3 | 1 | 128@112×112 | $128 \times (3 \times 3 \times 64 + 1)$ | $128 \times 112 \times 112$ |
| Conv2-2 | 128@112×112 | 3 × 3 | 1 | 128@112×112 | $128 \times (3 \times 3 \times 128 + 1)$ | $128 \times 112 \times 112$ |
| MaxPool_2 | 128@112×112 | 2 × 2 | 2 | 128@56×56 | 0 | $128 \times 56 \times 56$ |
| Conv3_1 | 128@56×56 | 3 × 3 | 1 | 256@56×56 | $256 \times (3 \times 3 \times 128+1)$ | $256 \times 56 \times 56$ |
| Conv3_2 | 256@56×56 | 3 × 3 | 1 | 256@56×56 | $256 \times (3 \times 3 \times 256 + 1)$ | $256 \times 56 \times 56$ |
| Conv3_3 | 256@56×56 | 3 × 3 | 1 | 256@56×56 | $256 \times (3 \times 3 \times 256 + 1)$ | $256 \times 56 \times 56$ |
| MaxPool_3 | 256@56×56 | 2 × 2 | 2 | 256@28×28 | 0 | $256 \times 28 \times 28$ |
| Conv4_1 | 256@28×28 | 3 × 3 | 1 | 512@28×28 | $512 \times (3 \times 3 \times 256 + 1)$ | $512 \times 28 \times 28$ |
| Conv4_2 | 512@28×28 | 3 × 3 | 1 | 512@28×28 | $512 \times (3 \times 3 \times 512 + 1)$ | $512 \times 28 \times 28$ |
| Conv4_3 | 512@28×28 | 3 × 3 | 1 | 512@28×28 | $512 \times (3 \times 3 \times 512 + 1)$ | $512 \times 28 \times 28$ |
| MaxPool_4 | 512@28×28 | 2 × 2 | 2 | 512@14×14 | 0 | $512 \times 14 \times 14$ |
| Conv5_1 | 512@14×14 | 3 × 3 | 1 | 512@14×14 | $512 \times (3 \times 3 \times 512 + 1)$ | $512 \times 14 \times 14$ |
| Conv5_2 | 512@14×14 | 3 × 3 | 1 | 512@14×14 | $512 \times (3 \times 3 \times 512 + 1)$ | $512 \times 14 \times 14$ |
| Conv5_3 | 512@14×14 | 3 × 3 | 1 | 512@14×14 | $512 \times (3 \times 3 \times 512 + 1)$ | $512 \times 14 \times 14$ |
| Deconv | 512@14×14 | 4 × 4 | 2 | 512@28×28 | $512 \times (4 \times 4 \times 512 + 1)$ | $512 \times 28 \times 28$ |
| Concat | 256@28×28, 512@28×28, 512@28×28 | | | 1280@28×28 | | $1280 \times 28 \times 28$ |
| Conv | 1280@28×28 | 1 × 1 | 1 | 512@28×28 | $512 \times (1 \times 1 \times 1280 + 1)$ | $512 \times 28 \times 28$ |

For the RPN part, the original VGG-based Faster-R-CNN chooses the output of the last convolutional layer for the region proposal. We operate the region proposal on the multiresolution feature map with a 2 times higher resolution. Moreover, considering the situation in which most of the ship ROIs are vertical after rotation and the shape of ships is narrow and long, we change the default anchor boxes aspect ratio from $1 \times 1$, $2 \times 1$, $1 \times 2$ to $1 \times 1$, $3 \times 1$, $1 \times 5$. These changes theoretically contribute to a higher precision for anchor locating and a higher IOU during the NMS procedure,

which is helpful to detect small ships and gathering ships. The whole network structure is shown in Figure 4.

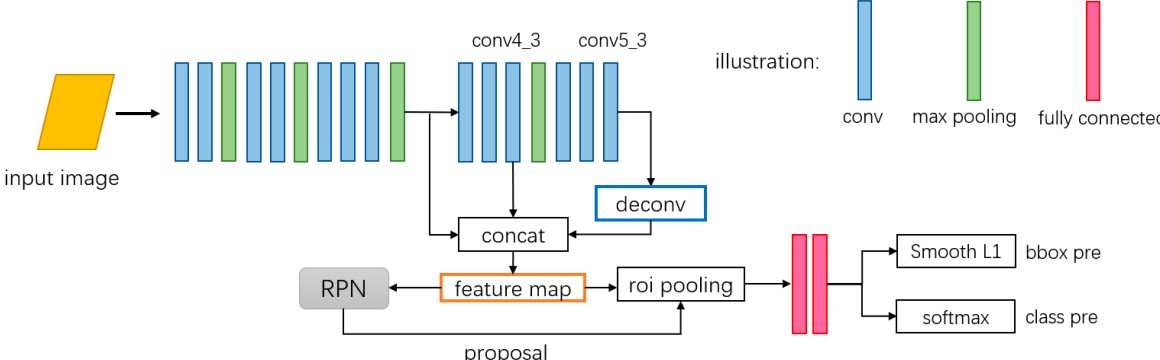

**Figure 4.** The structure of modified Faster-R-CNN.

## 2.3.2. Model Training and Result Process

There are two ways to train an R-CNN model. One is alternating optimization, which alternates training the CNN and RPN separately. Approximate joint training is another end-to-end way to train an R-CNN, which is used to train the modified VGG in this paper. Training the R-CNN model mainly includes the following processes: (1) prepare the dataset and divide the dataset into training samples and validation samples; (2) Set the parameters for model optimization; (3) Initialize the weights and bias for layers in the R-CNN model; (3) Input training data with labels, calculate the loss function, perform the error back propagation (BP); (4) Train the model until the value of the loss function is stable and converges for epochs; (5) If the value of the loss function does not converge or the results on the validation samples do not meet the requirements, adjust the parameters of the optimizer or the R-CNN model and retrain the model.

In the first step, we have 10 optical images of GF-2 and divide them into 2 parts, namely, 8 images for training and 2 images for validation and testing. We trained SVM first and obtained the rotated ship ROIs as mentioned previously. In addition, most of the ROI images contain ships in the vertical direction, while a few of the ships are not rotated in the assumed direction.

To enrich the training data and enhance the robustness of the model, data augmentation is necessary. Considering that there may be a few ships that are not rotated in the proper direction, we add both rotated and not rotated ROI images, and perform data augmentation on them. First, we augment the ROI images by randomly rotating the ROI in a range of degrees from $-30°$ to $30°$. Considering the diversity of the size and location of ships, we augment the ROI images by random image resizing and flipping vertically. We also add Gaussian noises and change the image illumination randomly. These methods of image augmentation are performed on every rotated training image so that the actual training data are 6 times larger of the original one. In detail, the data augmentation process is necessary because it ensures the diversity of training data and decreases the risk of overfitting.

We trained the model with a deep learning framework, Caffe. The optimizer for BP is the stochastic gradient descent (SGD) algorithm. A pretrained model that has been trained on the VOC dataset is used to initialize the weights and bias in the R-CNN model.

Once the model training is finished, we input ROI images to the R-CNN model and obtain ship bounding boxes as output results. The output of the model contains two parts: one is the image coordinates of the prediction box, (x, y, with height); another is the confidence score, which is the probability value of the existence of a ship in the prediction box. There may be a few boxes around the same ground truth box of a ship. To address this situation, the nonmaximum suppression (NMS) algorithm is used to select detection boxes with the highest confidence score. NMS uses intersection

over union (IoU) to eliminate the detection boxes around the same ground truth box. IoU can be calculated by Equation (1):

$$IoU = \frac{area(box1 \cap box2)}{area(box1 \cup box2)} \tag{1}$$

When the IoU of detection boxes is higher than a manually set threshold, NMS only maintains the detection box with the highest prediction confidence and drops the others.

## 3. Experiment and Results

### 3.1. Experiment Data

The experimental area is part of the Yangtze River in the rectangular area from (120.4°E, 32.0°N) to (121.7°E, 31.4°N). The map data is based on WGS-84 datum and UTM projection. The data we used are GF2 images whose resolution is 1 m after image fusion, and every one of these 12 images approximately covers a 27,000 m × 27,000 m area. There are not only ships with different sizes, types, and shapes, but also a large number of clustered ships. As shown in Figure 5, the approximate area covered by the training images is enclosed by dotted lines, and the test images are solid lines.

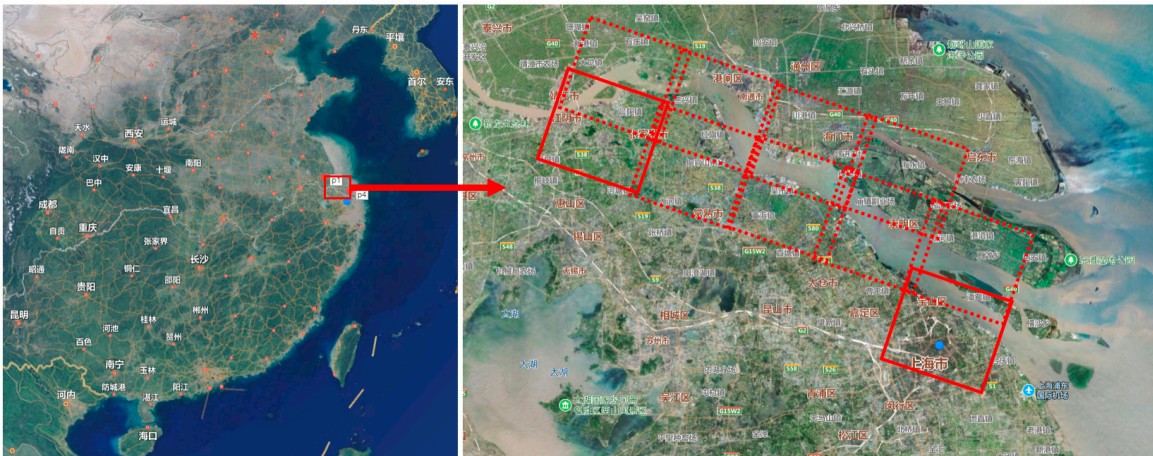

**Figure 5.** Approximate area covered by experiment data.

The experimental data consists of a training data part, including 8 remote sensing images and a validation data part including 2 images. The entire 4 bands of the remote sensing images were used to train SVM classifier to better distinguish the nonwater area from water area. However, only the band with the highest value was used to calculate the HOG features during the ship ROI extraction process. The two testing images are named image I and image II. Image I covers an area from (120°14′E, 32°1′N) to (120°32′E, 31°46′N), and image II covers an area from (121°24′E, 31°28′N) to (121°42′E, 31°14′N). The SVM classification results on the testing image are shown in Figures 6 and 7. In addition, the statistical results of the extraction of ship ROIs are counted manually and shown in Table 2.

To prepare the dataset for R-CNN model training, we select 1524 ship ROIs from the 8 training GF-2 images that contain 1705 ships as positive samples. There is no need to prepare negative samples for R-CNN. In addition, we used the ship ROIs detected by SVM to evaluate the model, which contains 595 image ROIs and 707 ships.

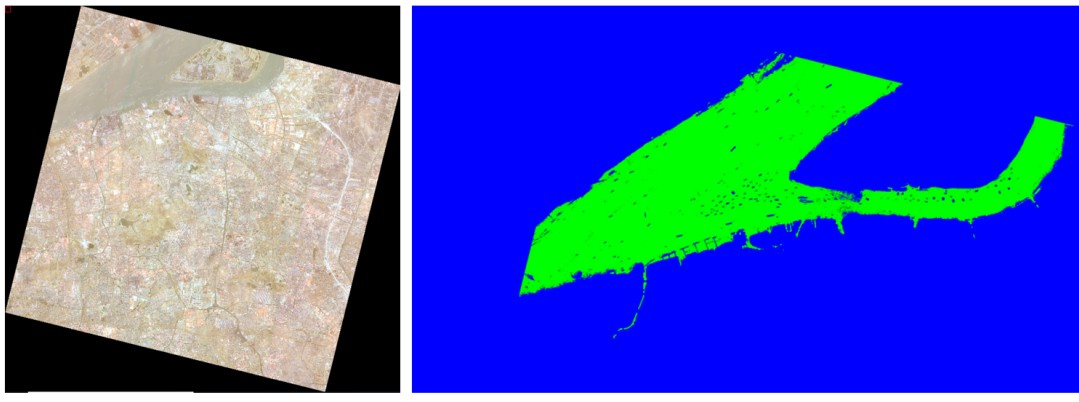

**Figure 6.** Water and nonwater classified result of test image I.

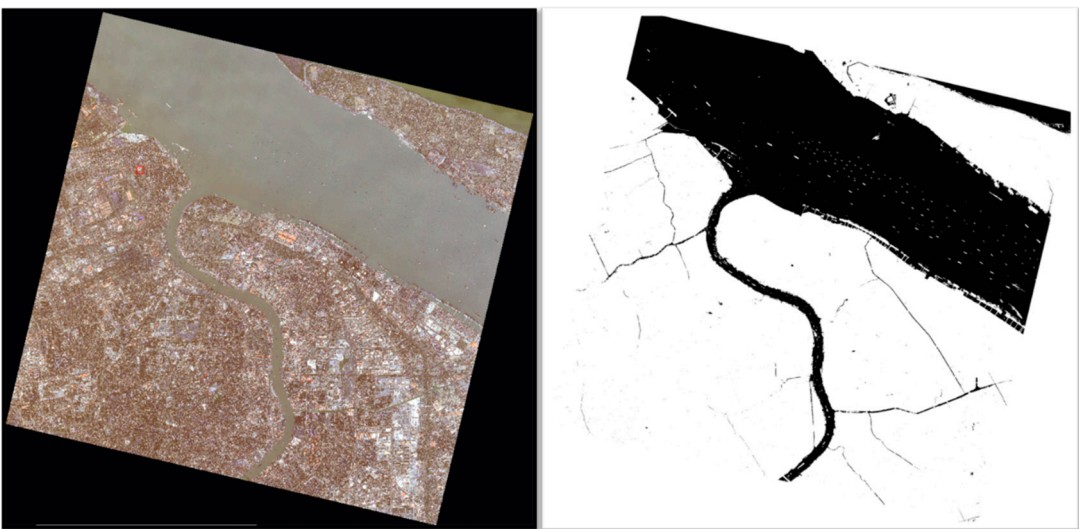

**Figure 7.** Water and nonwater classified result of test image II.

**Table 2.** The results of ship ROI extraction.

| Amount Statistic | Image I | Image II | Total |
|---|---|---|---|
| ships on inland river | 269 | 475 | 744 |
| ROIs | 238 | 357 | 595 |
| ships in ROIs | 254 | 453 | 707 |
| ships leaked from ROIs | 15 | 22 | 37 |
| ROIs contain no ship | 25 | 44 | 69 |
| rate of miss | 5.58% | 4.63% | 4.97% |
| rate of false alarm | 10.50% | 12.32% | 11.60% |

*3.2. Experimental Result*

We trained the mentioned R-CNN model by Caffe (a deep learning framework). The first three bands of the GF-2 images are used as RGB channels during model training. The initial learning rate is 0.001, the batch size of the input data is 2, and the momentum method and minibatch SGD are used for optimization.

We trained the original VGG-based Faster-R-CNN and our modified version by fine tuning on the pretrained VGG model provided by Caffe Model Zoo. This pretrained model was trained for the classifying task on ImageNet, which contains 14 million images and decreases the risk of overfitting. When training based on this model, most parameters in the neural network were able to begin with a reasonable initial value, which prompted the model to converge as soon as possible. At the beginning

of training, the modified VGG model had a higher loss and converged slower. After 30,000 iterations of BP, the loss of the two models tended to be the same. At the end of training for 50,000 iterations, the loss of the modified VGG model was slightly lower than that of VGG. The comparison of training details between the two models is shown in Figure 8.

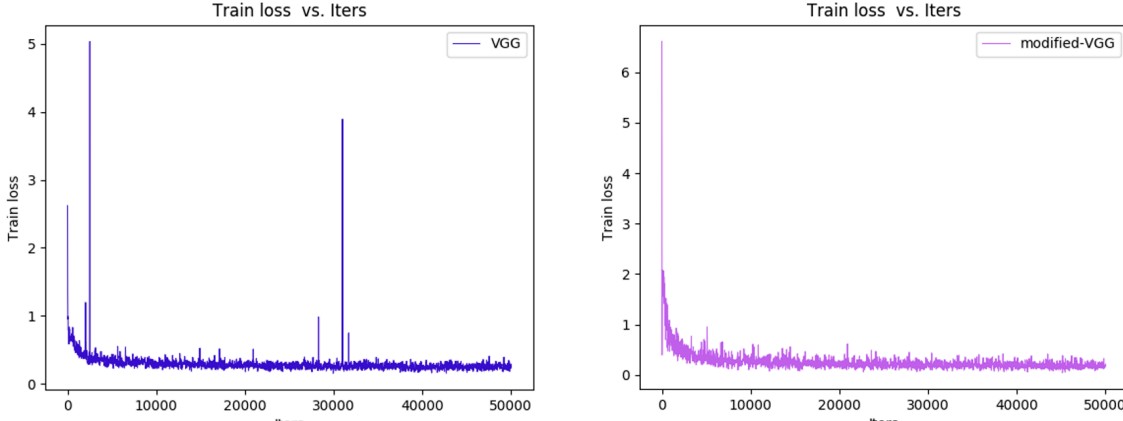

**Figure 8.** Training iterations, loss and accuracy curve.

For the model evaluation part, we evaluated the original VGG model and modified VGG model on the basis of the ship ROIs extraction result. There are 595 ROI images in total, 69 of them contain no ships as negative samples and the others contains 707 ships as positive samples.

For the result processing part, we use the NMS algorithm to process the result of the R-CNN outputs. To detect small and gathering ships, we need a low confidence threshold, which may cause some false detection boxes near the same ship. NMS calculates their IoU and maintains the detection box with the highest confidence. We set the confidence threshold and the IoU threshold in the NMS process after the R-CNN output to a lower and a higher value, which leads to different recall and precision; this is shown in Table 3.

**Table 3.** Comparison of ship detection results with different methods.

| Method | Confidence | IoU (NMS) | Recall | Precision |
|---|---|---|---|---|
| Faster-R-CNN (modified VGG16) | 0.4 | 0.3 | 96.46 | 95.79 |
| | 0.4 | 0.5 | 98.87 | 92.95 |
| | 0.5 | 0.3 | 93.21 | 97.20 |
| | 0.5 | 0.5 | 93.78 | 97.64 |
| Faster-R-CNN (original VGG16) | 0.4 | 0.3 | 94.06 | 95.27 |
| | 0.4 | 0.5 | 94.91 | 94.24 |
| | 0.5 | 0.3 | 91.65 | 95.72 |
| | 0.5 | 0.5 | 92.08 | 96.16 |
| Faster-R-CNN with ZF-net | 0.4 | 0.3 | 90.35 | 92.28 |
| SSD300 | 0.3 | 0.3 | 93.42 | 95.51 |
| YOLOv2-352 | 0.3 | 0.3 | 92.94 | 94.65 |
| DPM | | | 88.97 | 93.74 |

Finally, we conducted a comparative experiment. In this part, except for the original VGG-based Faster-RCNN and our improved Faster-RCNN, we implemented another 4 models, DPM, ZF-net based-Faster-RCNN, SSD300, and YOLOv2.

(1) To compare the effects between traditional detection methods and deep learning detection methods, we chose a most effective classic artificial feature-based method, DPM, and trained it with the same dataset implemented in R-CNN training. The DPM model is realized with HOG features and SVM, and each of the 6 submodels consist of 6 parts. The mixture of DPMs trained is shown in Figure 9.

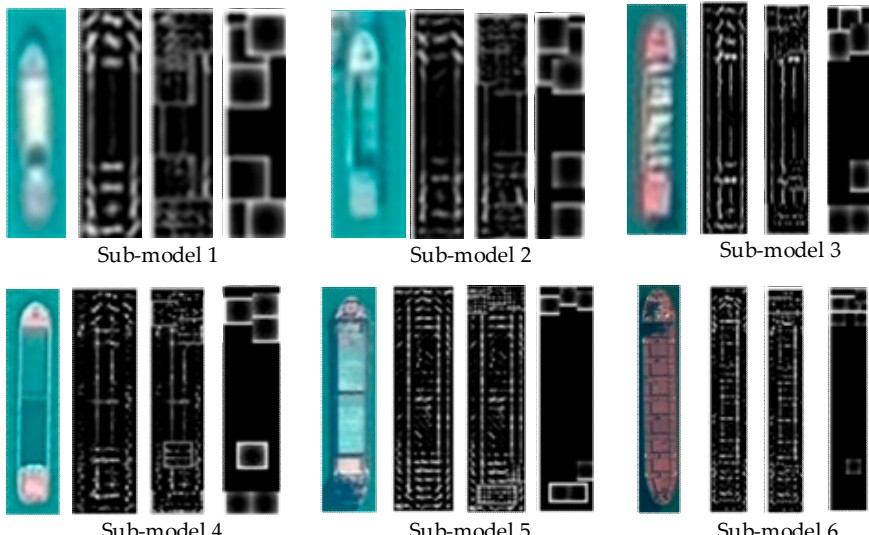

**Figure 9.** Mixture of deformable part models (DPMs) of ships.

(2) To compare the effects of image feature extraction and description between convolutional neural networks of different depth, we change the CNN part of Faster-R-CNN from VGG16 to ZF-net, which is a shallower CNN with only 5 convolutional layers. ZF-net downsamples the input image 16 times, which means the model input size is 244 × 244 and the output size is 13 × 13. It was trained with the same dataset, and the detection result can be found in Table 3.

(3) There are 3 major frameworks in the 2D target detection domain, R-CNN, SSD, and YOLO. To compare the effect of multisize ship detection between different deep learning target detection frameworks, we also trained a VGG-based SSD300 and YOLOv2 with the same dataset. Different from the Faster-RCNN framework, SSD does not have a professional network to provide box locating. It consists entirely of a convolutional network, and it also takes convolutional features of different depths into consideration. In fact, SSD has a loss function composed of 3 parts that originate from 3 detectors attached to the feature map with different resolutions, so that it is theoretically able to detect ships of different sizes. The 3 loss functions of SSD converge to a high loss value of approximately 1 to 2, and it needs a lower confidence threshold to maintain detection boxes. YOLOv2 has a different framework from both R-CNN and SSD. YOLO does not have RPN like R-CNN. It uses a cost function to fit the parameters of bounding box prediction directly that is similar to SSD. But there is no multiloss from different outputs of CNN designed in YOLO. Therefore, YOLO is predicted to be fast at calculation but less accurate. Finally, a comparison of the detection results in DPMs, SSD, original VGG-based R-CNN, and modified VGG-based R-CNN is shown in Table 3.

We compared the effect of detection results of the improved VGG-based Faster-RCNN under different confidence thresholds and IoU thresholds, and under situations such as ship images influenced by shadows and very blurry ships. This is shown in Figures 10 and 11.

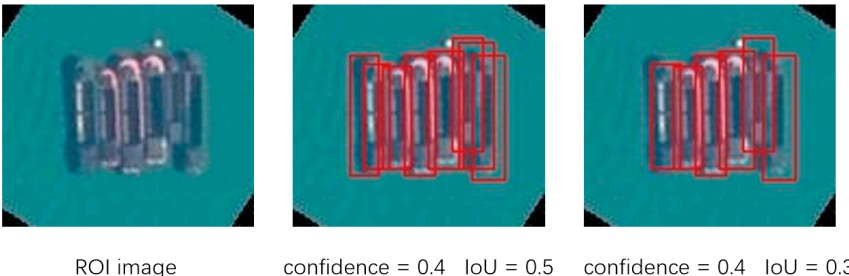

ROI image    confidence = 0.4   IoU = 0.5    confidence = 0.4   IoU = 0.3

**Figure 10.** The false detection caused by shadows can be eliminated with nonmaximum suppression (NMS).

We set the same confidence threshold (0.4) and IoU threshold (0.3) for ZF-net, VGG16, and our improved VGG16 because they share the same framework, while we set the confidence threshold to 0.3 and the IoU threshold to 0.3 for SSD300 and YOLO. Figure 12 shows the detection result on 2 testing images, including different imaging conditions, such as blurry ships, gathering ships, and ship clusters containing huge ships and small ships.

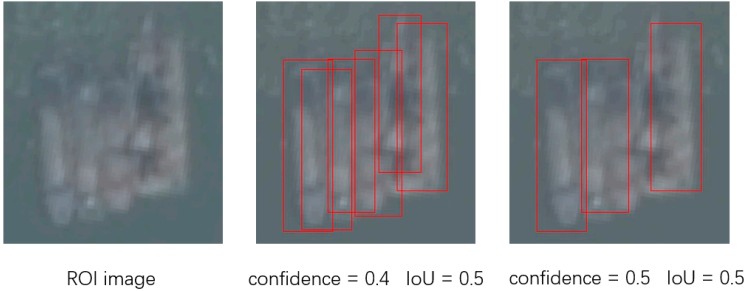

ROI image     confidence = 0.4  IoU = 0.5    confidence = 0.5  IoU = 0.5

**Figure 11.** The blurry ships can be detected with a lower confidence.

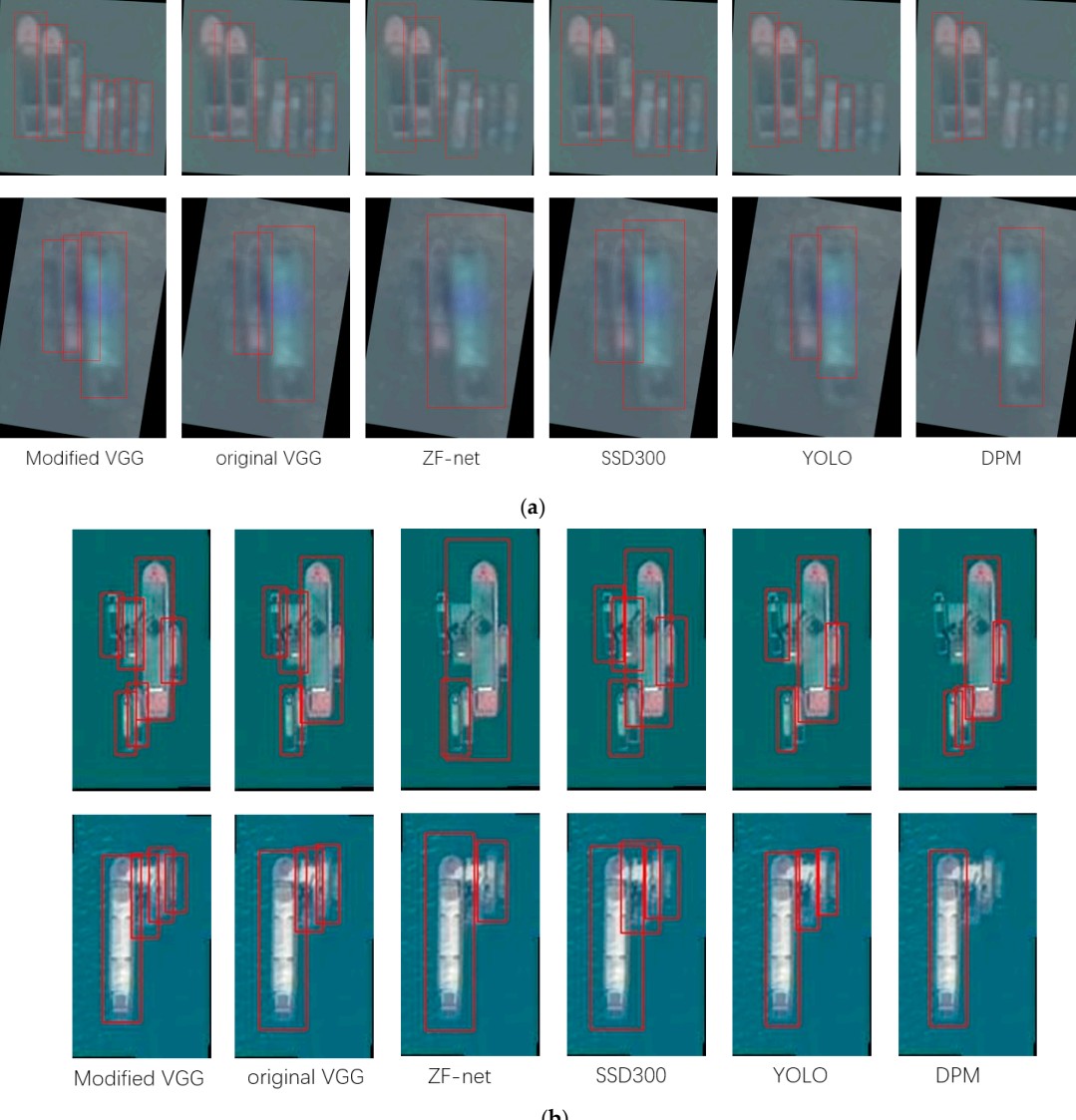

**Figure 12.** (**a**) Detection result on ROI from Image I; (**b**) Detection result on ROI from Image II.

Finally, we combine the results of image preprocessing, including SVM segmentation and ROI extraction (stage 1), and ship detection (stage 2). The final recall of our ship detection procedure can be calculated by Equation (2):

$$recall = \frac{ships\ detected\ rightly}{ships\ in\ test\ dataset} = recall\ of\ stage1 \times recall\ of\ stage2 \tag{2}$$

There are 707 ships extracted from a total of 744 ships in the preprocessing stage and the recall is 95.03%. The missing rate of this stage is 4.97%. Keeping the precision around 95%, as shown in Table 4, we can get the missing rate caused by the ship detection stage if we subtract the final recall from 95.03%. We find that if the normal RCNN method, SSD, YOLO, or DPM is used, the missing rate of the preprocessing stage is less than the detection stage. In this case, the performance of the whole detection procedure is mainly influenced by the second stage, so the precision and recall of the detection stage needs to be improved. Our improved RCNN method decreases the missing rate to a level less than the one caused by preprocessing stage. Once the recall of ship detection stage reaches 96%, the final recall is more evidently influenced by the preprocessing stage. It may be more economical to improve the recall of the preprocessing stage in further works.

**Table 4.** Missing rate analysis.

| Method | Precision | Stage Recall | Combine Recall | Missing Rate Caused |
|:---:|:---:|:---:|:---:|:---:|
| Preprocessing | | 95.03 | | 4.97 |
| Modified VGG | 95.79 | 96.46 | 91.66 | 3.36 |
| Original VGG | 95.27 | 94.06 | 89.39 | 5.64 |
| ZF-net | 92.28 | 90.35 | 85.86 | 9.17 |
| SSD300 | 95.51 | 93.42 | 88.78 | 6.25 |
| YOLOv2-352 | 94.65 | 92.94 | 88.32 | 6.71 |
| DPM | | 88.97 | 84.55 | 10.48 |

## 4. Discussion

From Table 3, we can observe that the recall of the R-CNN method, SSD, and YOLO is higher than that of the DPM method. With the exception of the ZF-net, other R-CNN methods, SSD, and YOLO achieve a higher precision than the DPM method. Though the precision of the modified VGG-based R-CNN is similar to DPM under a low confidence threshold, our model achieved a much higher recall. Furthermore, as shown in Figure 12, DPM obviously misses small ships and blurry ships. We can draw the conclusion that ship detection methods based on deep learning algorithms perform better in both recall and precision with a proper confidence threshold and NMS threshold than methods based on classic artificial image features and classifiers.

We find that a lower confidence can achieve a higher recall while the precision decreases at the same time. In the experiment, we set a very low confidence (0.1) threshold to print the classification scores of the top detections. We found that 0.4 is a value for the confidence threshold that is able to detect ships with low imaging quality. However, there are more false ship proposals detected in neighboring areas. When we set a lower IoU threshold in the NMS process, the false detection is eliminated and the precision increased. This situation is shown in Figure 10. When we set a confidence threshold to 0.5, there are less detections, but the detection boxes are more likely to contain ships. This is shown in Figure 11.

Moreover, for gathering ships and small ship detection, our modified VGG is capable of detecting gathering ships even when the image is sometimes blurry. We lower the confidence threshold to 0.4 and set the IoU threshold to 0.3 in NMS for R-CNN based on different CNN. As shown in Figure 12a, there are ships close to each other, and the image is not clear. Our modified VGG methods detect more ships than the original VGG and SSD, though the location is not very accurate. In Figure 12b, there are large ships and small ships joined together, and small ships are very close to each other. Our modified

VGG method is able to detect both large ships and small ships, though the IoU between the detection box and ground truth box can be low. Other methods can hardly distinguish gathering blurry ships or small ships from big ships.

## 5. Conclusions

In this paper, we proposed a ship detection method that is effective for small ships and gathering ships based on high-resolution remote sensing imagery. The method process adopted a coarse-to-fine strategy, in which segmentation of the nonwater area from the water area and extraction of the candidate areas that may contain ships (ROI) is performed first. Second, the R-CNN method is implemented to accurately detect ships in ROI images. Moreover, we improved the structure of the VGG16-based Faster-RCNN framework and made progress on the detection of small ships and gathering ships. Finally, in comparison experiments, our improved Faster-RCNN proved to achieve better recall and accuracy. In future work, we will try more traditional methods in the preprocessing stage to increase the recall of ROI, such as the variation of LBP, Gaussian Local Descriptors, SML, and PCA classifier. It will be worthwhile to conduct further research on more sophisticated CNN and even RNN methods based on optical remote sensing imagery.

**Author Contributions:** S.Z. and R.W. conceived, designed and performed the algorithm and experiments; K.X. performed major tunings in the parameters involved in the comparison experiments. J.W. and W.S. provided advice for the preparation and revision of the paper.

**Funding:** This research was funded by Science and Technology Commission of Shanghai Municipality, grant number 16DZ1100701 and The National Key Research and Development Program of China (2018YFB0505400).

**Acknowledgments:** The authors would like to thank the handling editor and anonymous reviewers for their reading and remarks.

**Conflicts of Interest:** The authors declare no conflict of interest.

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
