# Peer review of "R-CNN-Based Ship Detection from High Resolution Remote Sensing Imagery"

_remotesensing, doi:10.3390/rs11060631_

Round 1
Reviewer 1 Report
The authors propose a very interesting work based on CNN for ship detection.
The rationale of your full approach is clear and well explained.
You should run a more reliable testing protocol, e.g. run more time the split training and test, and then to report the average performance. Respect the original VGG you report better performance, but you tested your approach in a single dataset. Is your dataset & code available to download?
Since you have implemented other approaches, could you run tests combining the scores of SSD300 and/or FR-CNN with ZF-net with your approach?
In https://www.sciencedirect.com/science/article/pii/S2210832718301388 is shown that ensemble of CNN combined with other features permits to boost the performance. At least add some sentences as future works.
Author Response
Point 1: You should run a more reliable testing protocol, e.g. run more time the split training and test, and then to report the average performance. Respect the original VGG you report better performance, but you tested your approach in a single dataset. Is your dataset & code available to download?
Response 1: Thanks for advising and it’s a logistic suggestion that split training and testing and run more times, and analysis the average performance. On the one hand, the high-resolution remote sensing image data available for us is expensive and limited. We would split the training and testing dataset and do cross-validation, if we had high resolution remote sensing images from different area. And it can cost many days to train all the models mentioned in article. On the other hand, RCNN is a heavy model. So, we consider it’ better to use all train data that can be used while training, because making the most of data to train can decrease the risk of overfitting. We even use data augment to enrich the training data to achieve a better result.
Since the training data and the testing data are separated, the model is proved to be able to fit the data distribution on a certain area. When it comes to new tasks contains different area, the structure of neural network is stable, but the parameters of the model should be finetuned with new dataset.
The work was mainly completed separately by 3 authors. We put our source codes together to here https://pan.baidu.com/s/1WmH-Mm8YBFNC73FBXXQYFw. The dataset is available here https://pan.baidu.com/s/106LsHvbs3Z6NNmFP-aIdaA (passwords: “rcem”).
Point 2: Since you have implemented other approaches, could you run tests combining the scores of SSD300 and/or FR-CNN with ZF-net with your approach?
Response 2: We have combined the FR-CNN with ZF-net in our article and we use ZF-net mainly as a comparison experiment to analyse the performance of CNN with different depth under FR-CNN framework, especially to compare with our modified VGG under the same RCNN framework (section 3.2, paragraph 7, Table.3).
Point 3: In https://www.sciencedirect.com/science/article/pii/S2210832718301388 is shown that ensemble of CNN combined with other features permits to boost the performance. At least add some sentences as future works.
Response 3: We have read the articles you recommended, and been inspired by these works. The articles contained outstanding researches that applied deep learning algorithms to medical image domain. As suggested we added idea inspired by these works to future works in our article.
“A-survey-on-deep-learning-in-medical-image-analys_2017_Medical-Image-Analysis” comprehensively introduced the applications that deep learning methods developed for medical image analysis, including the CNN and RNN algorithm, supervised and unsupervised task, software and hardware and etc. It can be a reference for the development of deep learning algorithms in optical remote sensing image domain.
“Ensemble-of-texture-descriptors-and-classifiers-_2017_Applied-Computing-and-” included nearly all the best artificial image feature descriptions and SVM classifier. Though it is well known that the CNN classifier is more effective than artificial ones on a large dataset, the traditional descriptors and classifiers are still valuable in the application scenarios that are leaky of data or calculation ability. That’ s why we choose HOG and SVM to segment the non-water area from a large remote sensing image. And the article inspired us to try more traditional methods in pre-processing stage to increase the recall of ROI, such as LBPs, HASC, POEM, SML and PCA in future works.
In “Machine-Learning-Methods-for-Histopatho_2018_Computational-and-Structural-Bi” the process was similar to our work. We both need image segmentation and detection. While limited by hardware, the remote sensing image is too huge to perform deep learning segmentation methods at one time. The article noticed us the colour variation problem. To take the colour variation into consideration. We may add colour augmentation to data augmentation process, and operate colour normalization to pre-processing in future works.
“NiftyNet--a-deep-learning-platform-for-_2018_Computer-Methods-and-Programs-I” was a ambition work to build a deep learning platform for medical image applications. As a reference, we might build a platform for high resolution optical remote sensing image that provide a target detection pipline including data augmentation, segmentation and detection applications in the future.

Reviewer 2 Report
This paper proposes a ship detection method bases on Faster R-CNN for high resolution remote sensing image. In addition, some pre-processing methods, including SVM classification and morphology analysis, are added to guarantee the detection performance. The methodology is feasible, however, there are some issues should be improved before publishing in Remote Sensing.
1. There are many statements related to the ship detection in SAR image at the beginning of the manuscript. This would confuse readers since your method is not designed for SAR image.
2. Please enrich Section 2.2, especially the ROI generation.
3. Form the current version, only one operation is introduced to improve the Faster R-CNN, that is, the multiresolution image feature generation. The novelty is limited.
4. The experimental part is weak. First, the influence of the pre-processing methods should be discussed. Second, apart from the R-CNN related approaches, more comparisons should be added to prove the effectiveness of your algorithm.
5. Also in experimental section, it is well-known that R-CNN is a pretty heavy model for object detection. Compared with the natural scene, the volume of your training data (the number of objects and RoIs) is limited. How do you consider the overfitting issue during your training process?
Author Response
Point 1: There are many statements related to the ship detection in SAR image at the beginning of the manuscript. This would confuse readers since your method is not designed for SAR image.
Response 1: Thanks for pointing out the problem, we have simplified the introduction about SAR images in article. Considering plenty of researches on ship detection are based on SAR images, we remained a short introduction about ship detection with SAR images that leads to the following statements about researches on ship detection with optical remote sensing images (Introduction, paragraph 2).
Point 2: please enrich Section 2.2, especially the ROI generation
Response 2: We have added more details about how we extract ROI candidates from binary result image of SVM segmentation (Section 2.2). The main algorithms used in ROI extraction were realized with OpenCV, such as finding contours of foreground pixels, generating the minimum bounding rectangles of polygons and etc. Considering the main idea of this work is to provide a complete procedure to extract ships from a large remote sensing image and improve the structure of RCNN, we paid more attention to the RCNN part.
Point 3: from the current version, only one operation is introduced to improve the Faster R-CNN, that is, the multiresolution image feature generation. The novelty is limited.
Response 3: There are two major contribution in this article. The first one is to promote a coarse-to-fine strategy and a complete procedure to detect ships on a large remote sensing image. The second is that we applied RCNN method and improved it for small and gathering ships on high resolution optical remote sensing image. And small target detection is not easy for neural network 2D detection domain, especially for remote sensing image with normal resolution. The second contribution covered improving the structure of neural networks, data augment and network training. What’s more, we also trained a most effective traditional method (DPM) and other networks as comparison. And we hope this work is worth referring to other researches on ship detection.
Point 4: The experimental part is weak. First, the influence of the pre-processing methods should be discussed. Second, apart from the R-CNN related approaches, more comparisons should be added to prove the effectiveness of your algorithm
Response 4: First, we add statistics data and analysis to the result of the total procedure including pre-process part and ship detection part in article (section3.2 and Table.4). The major influence of the pre-processing to the whole ship detection procedure was discussed.
Second, we add another famous 2D target detection method different from R-CNN approaches to comparison. There are 3 major frameworks in 2D target detection domain, R-CNN, SSD and YOLO. We add YOLO to compare performances among different target detection frameworks. Therefore, we have taken CNNs of different depth, most popular 3 frameworks, and a mostly used traditional method (DPM) into comparison experiment. Every model could cost a few days to adjust the parameters for training and to fit the data distribution well. And the experiment has proved the effectiveness of our improved R-CNN by comparing with these representative methods mentioned above.
Point 5: Also in experimental section, it is well-known that R-CNN is a pretty heavy model for object detection. Compared with the natural scene, the volume of your training data (the number of objects and RoIs) is limited. How do you consider the overfitting issue during your training process?
Response 5: To overcome the overfitting problem in R-CNN training, we mainly adopted the following methods.
First, we did not train the model using random initial parameters which could cost many days to adjust training parameters and waiting for the cost function of network convergences. We used the pre-trained model, which was pre-trained for classify tasks on a large dataset (ImageNet) that contains 14 million images. We finetuned the network on our own dataset for ship detection task. Finetuning is a widely used way for CNN training now, which means the most of parameters in CNN has a good initial value. As suggested, we added introduction about finetuning on based of ImageNet in article (section 3.2, paragraph 2).
Second, limited to the data available, we did data augment to enrich the diversity of dataset. Data augment process is a quite normal process while training. As suggested, we added notes to point out the importance of data augment in article (section 2.3.2, paragraph 3). There is an opensource data augment tool written in Python, “imgaug”, that supplies many kinds of image transformation. We adopted flipping vertically, adding Gaussian noise, changing the light intensity, random resizing and rotation to enrich the data. So that the training data could increase to several times larger than original. What’s more the testing data is strictly separated from the training data. Therefore, the experimental results can prove the model is able to well fitted to the data of a certain area. For other detection task contains different areas, RCNN method should be finetuned on a new large enough dataset.

Round 2
Reviewer 1 Report
Accept in present form
Reviewer 2 Report
All of the issues are modified in the current version. I suggest that this version can be accepted.